# CAT: Circular-Convolutional Attention for Sub-Quadratic Transformers

**Yoshihiro Yamada**
Preferred Networks
yyamada@preferred.jp

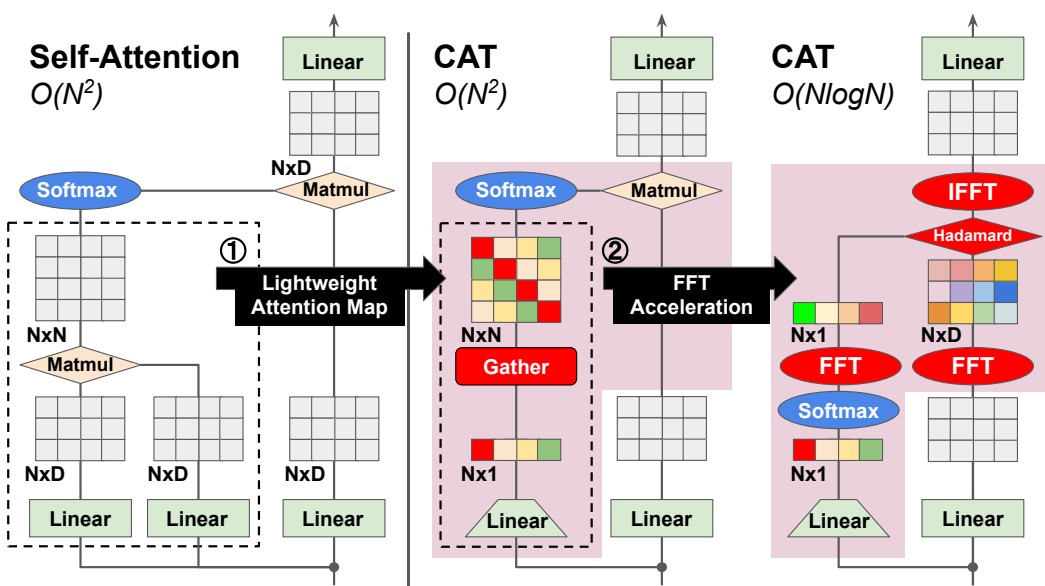

Figure 1: **From Self-Attention to CAT with two implementations.** Left: standard Self-Attention with a dense $N \times N$ attention map ($O(N^2)$). Middle: **CAT** ($O(N^2)$), a softmax-preserving circulant form of attention that reduces intermediate computations but remains quadratic overall. Right: **CAT** ($O(N \log N)$), the same circulant attention computed in the frequency domain using the Fast Fourier Transform (FFT), its inverse (IFFT), and an element-wise Hadamard product, achieving sub-quadratic complexity.

## Abstract

Transformers have driven remarkable breakthroughs in natural language processing and computer vision, yet their standard attention mechanism still imposes $O(N^2)$ complexity, hindering scalability to longer sequences. We introduce Circular-convolutional ATtention (CAT), a Fourier-based approach that efficiently applies circular convolutions to reduce complexity without sacrificing representational power. CAT achieves $O(N \log N)$ computations, requires fewer learnable parameters by streamlining fully connected layers, and introduces no additional heavy operations, resulting in consistent accuracy improvements and about a 10% speedup in naive PyTorch implementations. Based on the Engineering-Isomorphic Transformers (EITs) framework, CAT's design not only offers practical efficiency and ease of implementation, but also provides insights to guide the development of future high-performance Transformer architectures. Finally, our ablation studies

39th Conference on Neural Information Processing Systems (NeurIPS 2025).

highlight the key conditions underlying CAT's success, shedding light on broader principles for scalable attention mechanisms.

# 1   Introduction

Transformers have become the cornerstone of modern deep learning, excelling in natural language processing, computer vision, and beyond Vaswani et al. [2017], Kaplan et al. [2020], Dosovitskiy et al. [2021]. However, the $O(N^2)$ complexity of standard Self-Attention poses a formidable challenge for large-scale or real-world tasks Zhou et al. [2021], Wu et al. [2019], Liu et al. [2024]. To mitigate this, Linear Transformers attempt to reduce the complexity to $O(N)$ by adopting various kernel functions instead of softmax Katharopoulos et al. [2020], Choromanski et al. [2021]. Although these methods can handle long sequences, they often struggle to maintain the essential softmax-based weighting structure, leading to training instability and accuracy degradation Zhang et al. [2024]. Moreover, alternative architectures such as Mamba Gu and Dao [2023, 2024] deviate substantially from the original Transformer blueprint, increasing parameter counts and altering core mechanisms rather than serving as a true drop-in replacement Waleffe et al. [2024].

In this work, we introduce **Engineering-Isomorphic Transformers (EITs)**, a class of attention-based models that must retain the core softmax weighting while achieving sub-quadratic time. This concept helps preserve the strong representational power of standard attention, avoiding many of the pitfalls (e.g., numerical instability, partial token coverage) seen in previous approximations. As a concrete instantiation, we propose **Circular-convolutional ATtention (CAT)**, which leverages Fast Fourier Transform (FFT) and its inverse (IFFT) based circular convolutions to reduce the naive $O(N^2)$ cost to $O(N \log N)$. Unlike kernel or sparse approximations, CAT maintains a global softmax and does not introduce extra hyperparameters tied to the sequence length. We validate CAT on large-scale vision (ImageNet-1k) and language (WikiText-103) tasks, demonstrating consistent speedups over standard attention and comparable or improved accuracy. In addition, an ablation study reveals key design factors, such as merging query and key projections, that enable CAT to serve as a drop-in replacement under the EIT principles.

Our main contributions are as follows.

- **EITs: A new framework.** We formally define *Engineering-Isomorphic Transformers* (EITs), a class of sub-quadratic yet fully softmax-preserving architectures that clarifies how efficiency and the softmax-based weighting form can coexist.

- **CAT: FFT-based attention.** We propose *Circular-convolutional ATtention* (CAT), which uses FFT-based circular convolutions to reduce the naive $O(N^2)$ cost to $O(N \log N)$, while maintaining global softmax behavior and requiring fewer parameters, matrix operations, and no hyperparameters.

- **Empirical validation.** On ImageNet-1k and WikiText-103, CAT consistently matches or exceeds standard attention under simpler token mixing (e.g. average pooling, masked inputs), providing speedup in naive implementations. Looking ahead, we believe CAT's sub-quadratic design, solidly grounded in EIT principles, opens new directions for *longer-sequence* modeling across language and vision tasks. Combining CAT with other efficient attention mechanisms or advanced GPU kernels may unlock even greater scalability in future Transformer architectures.

# 2   Engineering-Isomorphic Transformers (EITs)

**Standard Self-Attention Recap.** Let $\mathbf{X} \in \mathbb{R}^{N \times D}$ be an input sequence of length $N$ and feature dimension $D$. A standard Transformer projects $\mathbf{X}$ into queries, keys, and values:

$$\mathbf{Q} = \mathbf{X}\mathbf{W_Q}, \quad \mathbf{K} = \mathbf{X}\mathbf{W_K}, \quad \mathbf{V} = \mathbf{X}\mathbf{W_V},$$

where $\mathbf{W_Q}, \mathbf{W_K}, \mathbf{W_V} \in \mathbb{R}^{D \times D}$. Then, the Self-Attention is computed as

$$\text{Attention}(\mathbf{Q}, \mathbf{K}, \mathbf{V}) = \text{softmax}\left(\frac{\mathbf{Q}\mathbf{K}^\top}{\sqrt{D_k}}\right)\mathbf{V} \in \mathbb{R}^{N \times D},$$

where $\mathrm{softmax}$ is applied row-wise, $D_k$ is the key/query dimension, and each row in the resulting $N \times N$ matrix sums to 1.

**Engineering-Isomorphic Transformers.** We now formalize a class of Transformers that preserve the softmax-based weighting form of standard Self-Attention while achieving sub-quadratic complexity. We refer to such models as **Engineering-Isomorphic Transformers (EITs)**. The notion of EITs helps us reason about efficiency and representational fidelity in a unified framework, offering clear guidelines for designing novel attention mechanisms.

Formally, EITs must satisfy four requirements:

1. **Softmax Preservation.** There exist functions $\mathrm{F_{attn}}(\mathbf{X})$ and $\mathrm{F_{value}}(\mathbf{X})$ such that the core attention mechanism can be written as

$$\mathrm{F_{out}}(\mathbf{X}) = \mathrm{softmax}\big(\mathrm{F_{attn}}(\mathbf{X})\big)\,\mathrm{F_{value}}(\mathbf{X}),$$

   mirroring $\mathbf{Q}\mathbf{K}^\top$ based weighting and row-wise normalization in standard attention. This ensures the global context dependencies remain intact.

2. **Sub-Quadratic Complexity.** $\mathrm{F_{out}}(\mathbf{X})$ must be computable in strictly less than $O(N^2)$ time (where $N$ is the sequence length).

3. **Parameter Efficiency.** The total number of trainable parameters should remain comparable to (or smaller than) standard multi-head attention.

4. **Minimal Hyperparameter Overhead.** No sequence-length-dependent hyperparameters that require careful tuning (e.g., block sizes, custom sparsity patterns) should be introduced.

**Design space.** EITs are not tied to a single construction. In general, any mechanism that retains the softmax-based weighting form of standard attention while executing the attention computation in time below quadratic (e.g., $O(N \log N)$ or $O(N)$) falls within the EIT design space. Examples include value-stream reductions that keep a global row-wise softmax (e.g., average or learned pooling prior to value mixing), biasing or Hadamard gating that does not alter the row-wise normalization, and other Toeplitz/circulant-like transforms. CAT should be viewed as one concrete instance in this broader space, rather than the only one.

**Why preserve softmax?** We do not claim that exact softmax is universally optimal; efficiency, accuracy, and theory ultimately decide. Still, keeping softmax is a pragmatic axis to explore: it maintains normalized and interpretable token weights, remains compatible with established training stacks and inference kernels (e.g., Flash-style implementations, KV caching, cross-attention), and avoids token-subset coverage effects. This axis complements kernel/linear approaches that discard softmax and search a different approximation space: EITs keep the softmax form while seeking implementations below quadratic time. Our results with CAT indicate that this branch can be made efficient (e.g., an $O(N \log N)$ realization with favorable wall-clock and memory profiles) without sacrificing accuracy.

## 2.1 Properties and Theoretical Insights

In defining EITs, two key challenges must be addressed to ensure that $\mathrm{F_{out}}(\mathbf{X})$ meets sub-quadratic requirements and remains practically viable. First, we need to establish theoretical efficiency, the attention operation must truly circumvent the $O(N^2)$ bottleneck. Second, even if we achieve a lower computational complexity on paper, the method must exhibit practical performance validity when scaled to real-world applications.

**Theoretical Efficiency** A core requirement of EITs is the capacity to compute $\mathrm{F_{out}}(\mathbf{X})$ in strictly less than $O(N^2)$ time. Naively multiplying an $N \times N$ matrix by an $N \times D$ matrix would ordinarily incur $O(N^2)$ complexity, particularly if $\mathrm{F_{attn}}(\mathbf{X})$ directly corresponds to $\mathbf{Q}\mathbf{K}^\top$. To circumvent this, one must either avoid explicitly constructing the full attention matrix or leverage specialized transforms that reduce the overall cost. For instance, in Sec. 4 we introduce CAT, which exploits Fourier-based circular convolutions to achieve $O(N \log N)$ complexity Cooley and Tukey [1965], all while retaining the softmax-based weighting structure central to Self-Attention.

**Practical Performance Validity** However, merely satisfying sub-quadratic complexity does not guarantee strong empirical performance. Large-scale tasks in language processing or vision demand

not just efficient computations but also robust training stability and high accuracy. Many existing approximations, though efficient in principle, struggle with degraded performance when confronted by real-world data scales. Thus, practical performance validity becomes essential: an EIT must demonstrate that its theoretical gains do not come at the expense of representational capacity or overall results. In Sec.5, we present empirical evidence verifying that our approach preserves (and often improves) the performance of standard Transformers, illustrating that a sub-quadratic softmax-based mechanism can indeed align with practical, large-scale demands.

# 3 Related Work

Having established the concept of EITs, we now evaluate how widely used attention-reduction techniques align with or diverge from this framework.

Approaches like BigBird Zaheer et al. [2020] and Longformer Beltagy et al. [2020] reduce attention to specific regions or compressed representations, effectively pushing complexity below $O(N^2)$. While such strategies preserve a row-wise softmax, it is only applied to a subset of tokens, thereby breaking the *global* weighting that we consider essential for EITs. Moreover, many sparse or low-rank architectures rely on carefully tuned patterns (e.g., block sizes, sparsity thresholds) that may need readjustment as sequence length $N$ grows, introducing additional hyperparameter overhead. This conflicts with the minimal-hyperparameter principle we outlined above.

Performer Choromanski et al. [2021] and Linear Transformers Katharopoulos et al. [2020] introduce kernel-based feature mappings to approximate softmax. Although this yields $O(N)$ scaling, the exact softmax structure is lost, which may affect training stability and interpretability Zhang et al. [2024].

Methods such as S4 Gu et al. [2022] or Mamba Gu and Dao [2023, 2024] eschew attention entirely, employing continuous-time or RNN-like mechanisms. While efficient, they diverge from the fundamental idea of a data-dependent softmax weighting over all pairs of tokens.

By contrast, EITs preserve a global softmax mechanism and still operate with sub-quadratic complexity. In Sec. 4, we introduce *CAT* as a concrete instantiation of this principle, illustrating how a Fourier-based approach can retain the core benefits of attention while breaking the $O(N^2)$ barrier.

# 4 Proposed Method

We propose **Circular-convolutional ATtention (CAT)**, an approach that meets the EITs criteria (Sec. 2). CAT applies circular convolutions in the frequency domain to reduce the $O(N^2)$ cost of Self-Attention to $O(N \log N)$, while preserving the essential global softmax structure.

**Key Idea.** Rather than explicitly computing $\mathrm{softmax}(\mathbf{Q}\mathbf{K}^\top)$, CAT interprets the attention weights as a *circulant* (or *circular shift*) matrix. Specifically, we learn a single projection matrix $\mathbf{W_A} \in \mathbb{R}^{D \times 1}$ to map the input $\mathbf{X} \in \mathbb{R}^{N \times D}$ into a vector

$$\mathbf{Z} = \mathbf{X}\mathbf{W_A} \in \mathbb{R}^{N \times 1}.$$

We then apply a row-wise softmax, yielding $\mathbf{Z}^\star = \mathrm{softmax}(\mathbf{Z})$. This $\mathbf{Z}^\star$ serves as the first row of a circulant matrix $\mathrm{circ}(\mathbf{Z}^\star)$, thus representing global pairwise interactions. In particular, we define

$$\mathrm{circ}(\mathbf{Z}^\star) = \begin{bmatrix} Z_1^\star & Z_2^\star & \dots & Z_N^\star \\ Z_N^\star & Z_1^\star & \dots & Z_{N-1}^\star \\ \vdots & \vdots & \ddots & \vdots \\ Z_2^\star & Z_3^\star & \dots & Z_1^\star \end{bmatrix},$$

so each row is a one-step circular shift of the previous row. Convolving this matrix with $\mathbf{X}\mathbf{W_V}$ (where $\mathbf{W_V} \in \mathbb{R}^{D \times D}$) can be performed via FFT, thereby avoiding the naive $N \times N$ multiplication.

Formally, let $\mathbf{V} = \mathbf{X}\mathbf{W_V}$. Then,

$$\mathbf{F}_{\mathrm{cat}} = \mathrm{circ}(\mathbf{Z}^\star)\mathbf{V} \quad \Longleftrightarrow \quad \mathrm{IFFT}\big[\mathrm{FFT}(\mathbf{Z}^\star) \odot \mathrm{FFT}(\mathbf{V})\big],$$

where $\odot$ denotes the Hadamard (element-wise) product applied in the frequency domain, and the overall computation is performed in $O(N \log N)$ time.

Unlike kernel-based approximations, this FFT-based procedure does not distort the softmax distribution, since the circulant structure is exactly equivalent to circular convolution (up to floating-point rounding). Qualitative attention-map analyses in Appendix A are consistent with this interpretation, showing structured, shift-like patterns induced by the circulant design.

**Multi-head Extension.** CAT naturally extends to multi-head attention by maintaining separate sets of $(\mathbf{W_A}, \mathbf{W_V})$ per head. Except for this, no additional modifications are needed. We refer readers to Sec. 5 for performance results in image classification and language modeling. Moreover, CAT naturally integrates typical attention optimizations, such as key-value caching during inference.

## 4.1 Implementation Details

We consider two ways of implementing CAT. The *gather-based* version, while nominally $O(N^2)$ due to indexing overhead, consistently yields about a 10% speedup of each iteration in naive PyTorch. Complete runtime and memory profiles are provided in Appendix B. In contrast, a *full FFT-based* version can theoretically achieve $O(N \log N)$ complexity, although we currently observe minimal gains at moderate $N$. We detail these comparisons below.

**Gather-based Approach.** By rolling the value matrix $\mathbf{V}$ via `torch.gather`, we avoid large matrix multiplications. Despite having an $O(N^2)$ indexing, we see a net 10% speedup over standard attention (e.g., for ViT CLIP-L on NVIDIA V100 GPUs). We attribute this to simpler dataflows and fewer partial matrix operations.

**FFT-based Approach.** Applying FFT/IFFT directly to $\mathbf{Z}^\star$ and $\mathbf{V}$ realizes the sub-quadratic $O(N \log N)$ complexity. However, for $N = 256$, the overhead of FFT calls (and their GPU kernels) often offsets the theoretical advantage. We anticipate greater speedups for larger $N$ and specialized GPU FFT implementations, which we leave for future work.

## 4.2 Key Theoretical Considerations and Advantages

Beyond achieving sub-quadratic complexity, our CAT design offers several theoretical merits that ensure it remains faithful to the core properties of Self-Attention. We briefly summarize these points here.

**Minimal Row-Wise Softmax Structure.** A key insight is that CAT maintains an *exact* row-wise softmax normalization by leveraging circulant matrices. Formally, for any $\mathbf{Z}$,

$$\mathrm{softmax}(\mathrm{circ}(\mathbf{Z})) \equiv \mathrm{circ}(\mathrm{softmax}(\mathbf{Z})),$$

since both the circulant operator and the softmax function act independently on rows. Thus, CAT meets the strict EIT requirement of a global row-normalized weighting, providing a "minimal" row-wise transform that does not distort the softmax attention. Introducing additional transformations would typically break this row-wise property and lose the exact softmax coverage.

**Dramatically Reduced Attention-Map Materialization.** Unlike standard $\mathbf{Q}\mathbf{K}^\top$ attention, which computes an $N \times N$ matrix of attention scores at runtime, CAT represents attention with a length-$N$ circulant kernel $\mathbf{Z}^\star$ derived from $\mathbf{X}\mathbf{W_A}$, from which the full attention matrix is generated by circular shifts. This reduces the number of attention coefficients that must be produced and operated on from $N^2$ to $N$ (per head), lowering intermediate compute and memory traffic and enabling $O(N \log N)$ FFT-based execution. We observe that this structured reduction can stabilize optimization and, in some configurations, improve generalization (e.g., Tab. 3).

**Explicit Relative Positioning.** In many tasks, the relative ordering of tokens (e.g. word positions, time steps, or spatial locations) plays a vital role. While classical Transformers rely on positional embeddings to hint at absolute positions, they lack a dedicated mechanism for handling relative offsets. Empirical studies of early Transformer layers Clark et al. [2019] suggest attention maps often exhibit shift-like or local patterns. By explicitly enforcing a circulant structure, CAT encodes this shift-based symmetry from the outset. Hence, early layers can more naturally capture local recurrences or repeated patterns in the data, which may lead to faster convergence and improved feature extraction.

**Complementarity with Standard Attention.** Despite these advantages, our experiments (Sec. 5) show cases where standard Self-Attention still excels, especially if highly flexible global interactions

| Model | Pool Type | Mechanism | Learnable | Complexity | Acc.↑ |
|---|---|---|---|---|---|
| CLIP-B | token | Attention | $3D^2$ | $O(N^2)$ | 0.574 |
| CLIP-B | token | CAT (ours) | $(D+H)D$ | $O(N \log N)$ | 0.540 |
| CLIP-B | token | CAT-Alter (ours) | $(2D+H/2)D$ | $O(N^2)$ | **0.582** |
| CLIP-L | token | Attention | $3D^2$ | $O(N^2)$ | 0.574 |
| CLIP-L | token | CAT (ours) | $(D+H)D$ | $O(N \log N)$ | 0.559 |
| CLIP-L | token | CAT-Alter (ours) | $(2D+H/2)D$ | $O(N^2)$ | **0.593** |
| CLIP-B | avg | Attention | $3D^2$ | $O(N^2)$ | 0.638 |
| CLIP-B | avg | CAT (ours) | $(D+H)D$ | $O(N \log N)$ | 0.649 |
| CLIP-B | avg | CAT-Alter (ours) | $(2D+H/2)D$ | $O(N^2)$ | **0.662** |
| CLIP-L | avg | Attention | $3D^2$ | $O(N^2)$ | 0.646 |
| CLIP-L | avg | CAT (ours) | $(D+H)D$ | $O(N \log N)$ | **0.694** |
| CLIP-L | avg | CAT-Alter (ours) | $(2D+H/2)D$ | $O(N^2)$ | 0.681 |

Table 1: ImageNet-1k results on ViT CLIP-B/L with different pooling strategies. Here, $D$ is the input embedding dimension, and $H$ is the number of attention heads. Recent optimizations can reduce this overhead, but are not considered in our baseline. CAT excels when the token mixing is simpler (e.g., avg), while CAT-Alter is competitive or superior across most settings.

| Model | LM Type | Mechanism | Learnable | Complexity | Word PPL↓ |
|---|---|---|---|---|---|
| Transformer-XL | masked | Attention | $3D^2$ | $O(N^2)$ | 13.94 |
| Transformer-XL | masked | CAT (ours) | $(D+H)D$ | $O(N \log N)$ | 10.28 |
| Transformer-XL | masked | CAT-Alter (ours) | $(2D+H/2)D$ | $O(N^2)$ | **8.51** |
| GPT-2 small | masked | Attention | $3D^2$ | $O(N^2)$ | 9.82 |
| GPT-2 small | masked | CAT (ours) | $(D+H)D$ | $O(N \log N)$ | 8.32 |
| GPT-2 small | masked | CAT-Alter (ours) | $(2D+H/2)D$ | $O(N^2)$ | **7.54** |
| Transformer-XL | causal | Attention | $3D^2$ | $O(N^2)$ | **30.82** |
| Transformer-XL | causal | CAT (ours) | $(D+H)D$ | $O(N^2)$ | 36.71 |
| Transformer-XL | causal | CAT-Alter (ours) | $(2D+H/2)D$ | $O(N^2)$ | 30.93 |
| GPT-2 small | causal | Attention | $3D^2$ | $O(N^2)$ | 27.84 |
| GPT-2 small | causal | CAT (ours) | $(D+H)D$ | $O(N^2)$ | 32.36 |
| GPT-2 small | causal | CAT-Alter (ours) | $(2D+H/2)D$ | $O(N^2)$ | **27.68** |

Table 2: WikiText-103 (masked and causal language modeling). Here, $D$ is the input embedding dimension, and $H$ is the number of attention heads. CAT shows notable gains in the masked setting, while CAT-Alter remains more robust in the causal setup.

are required. However, combining CAT and standard attention within a single network (e.g., CAT-Alter) can yield complementary benefits, as CAT enforces shift-based regularization while standard attention remains fully expressive. Thus, even if CAT alone is occasionally suboptimal, this hybrid approach can surpass both purely circulant and purely attention-based methods, offering a practical balance between efficiency and accuracy.

Overall, CAT embodies a "least complex" row-wise transform that fully preserves the softmax mechanism while drastically reducing the attention overhead. In the following sections, we detail how these properties translate into strong empirical performance, especially under simpler token mixing (*avg* pooling) or masked language modeling regimes (Sec. 5).

## 5 Experiments

We evaluate our CAT on two major benchmarks: ImageNet-1k Russakovsky et al. [2015] for image classification and WikiText-103 Merity et al. [2016] for language modeling. Our main focus is to demonstrate (1) whether CAT can maintain or improve performance despite its sub-quadratic design, and (2) how a hybrid variant that partially retains standard attention might further boost robustness.

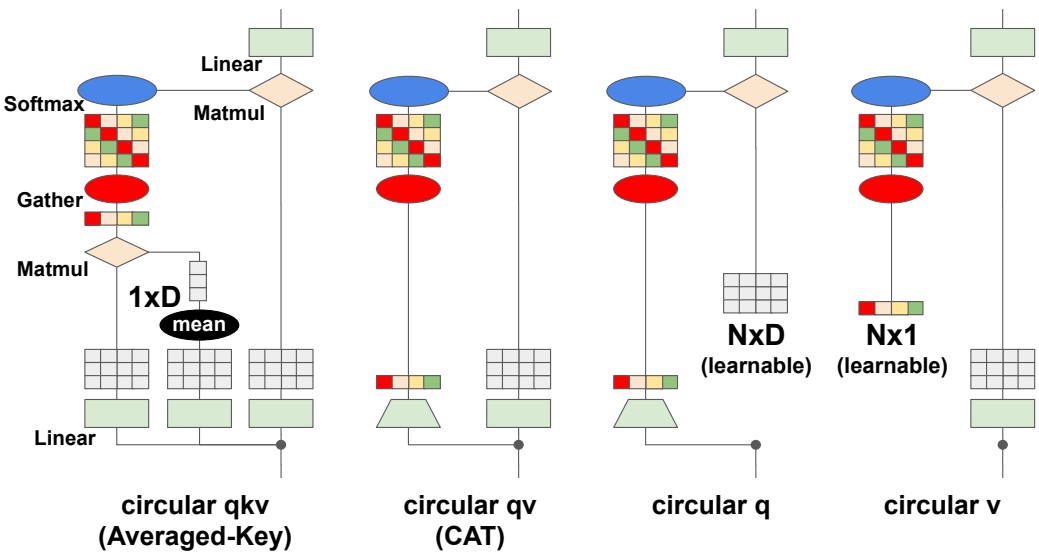

Figure 2: **Ablation study comparing different parameterization strategies for query, key, and value (qkv, qv, q, v).** Although fully splitting qkv (Averaged-Key) can yield slightly higher accuracy, it reintroduces attention-level parameter budgets. Our $qv$ variant (CAT) strikes a practical balance, maintaining sub-quadratic complexity and competitive performance.

| Model | Pool Type | Mechanism | Circular qkv | Learnable | Complexity | Acc.↑ |
|-------|-----------|-----------|--------------|-----------|------------|-------|
| CLIP-L | avg | Attention | - | $3D^2$ | $O(N^2)$ | 0.646 |
| CLIP-L | avg | Circular | qkv (Averaged-Key) | $3D^2$ | $O(N \log N)$ | **0.696** |
| CLIP-L | avg | Circular | qv (CAT) | $(D+H)D$ | $O(N \log N)$ | **0.694** |
| CLIP-L | avg | Circular | q | $(N+H)D$ | $O(N \log N)$ | 0.637 |
| CLIP-L | avg | Circular | v | $(N+D)D$ | $O(N \log N)$ | 0.625 |

Table 3: Ablation on key-value parameterization under circular convolution. Here, $D$ is the input embedding dimension, $H$ is the number of attention heads, and $N$ is the input time series. While qkv achieves slightly higher accuracy, qv (CAT) remains competitive with fewer parameters.

We primarily use ViT CLIP-B/L Radford et al. [2021], Cherti et al. [2023] for image classification and a Transformer-XL Dai et al. [2019] / GPT-2 small Radford et al. [2019] backbone for language modeling. We follow the original architectures for CLIP-B (12 heads), CLIP-L (16 heads), Transformer-XL (10 heads), and GPT-2 small (12 heads) preserving the same number of attention heads in both standard and CAT layers. Training hyperparameters, such as learning rate and batch size, are likewise inherited from each respective baseline. Where indicated, we replace all attention layers with CAT (denoted **CAT**) or half of them (denoted **CAT-Alter**), while the rest remain standard attention. In the latter case, the network interleaves CAT and standard attention blocks. Although half the layers still run in $O(N^2)$, the other half operate in $O(N \log N)$ via CAT, yielding a net speedup in practice. Moreover, because CAT merges query and key projections, these replaced layers reduce the overall number of learnable parameters relative to fully attentive models.

Our comparisons include the original Transformer attention for each architecture. We report standard validation accuracy on ImageNet-1k and validation word perplexity (word PPL) on WikiText-103. Complete runtime and memory profiles are provided in Appendix B.

## 5.1 Training Details

We train our models on the ImageNet-1k training dataset Russakovsky et al. [2015] using a batch size of 256 and a standard input resolution of $224 \times 224$ on 4 NVIDIA V100 GPUs. The initial learning rate is set to $2 \times 10^{-5}$, with weight decay of $1 \times 10^{-4}$. All models are randomly initialized. We train

for 50 epochs, applying a 10-epoch warmup phase followed by a cosine-annealing scheduler. We use *AdamW* with default hyperparameters (i.e., $\beta_1 = 0.9$, $\beta_2 = 0.999$). Data augmentation consists of random cropping and horizontal flipping.

For language modeling on the WikiText-103 training dataset Merity et al. [2016], we train with a batch size of 128 and an initial learning rate of $2.5 \times 10^{-4}$ on 4 NVIDIA V100 GPUs. We run 50 total epochs, employing a 1000-iteration warmup. We set the maximum sequence length to 256. A dropout rate of 0.1 is applied, and gradient norms are clipped at a maximum of 0.25. As with ImageNet-1k, we use *AdamW* Loshchilov and Hutter [2019] under default settings unless stated otherwise. Models are also randomly initialized in this setup. Finally, for masked language modeling experiments, we use a masking probability of 0.15.

In causal language modeling, we shift **Z** to ensure each position attends only up to its current timestep. However, enforcing strict causality reintroduces computational overhead, typically reverting CAT's complexity to $O(N^2)$ via explicit masking or repeated circular shifts. Fully sub-quadratic implementations under causal constraints remain an important open challenge.

## 5.2 Experimental Results

We examine image classification tasks on ImageNet-1k (Tab. 1). We test two pooling strategies: **token** pooling via a special classification token, and **avg** pooling over the entire sequence. Overall, **CAT** tends to excel under *simpler token mixing*, as seen with average pooling, while **CAT-Alter** appears more robust across different setups. Notably, CAT-Alter can outperform standard attention in several configurations, illustrating that a partial replacement can deliver improvements without fully discarding the attention mechanism.

We further examine masked and causal language modeling on WikiText-103 (Tab. 2). **CAT** attains strong gains in masked modeling scenarios, suggesting that its sub-quadratic structure remains highly effective when masking or simpler token manipulations are involved. In contrast, **CAT-Alter** often achieves performance close to, or slightly better than, standard attention across a wider range of conditions, indicating that partial adoption of CAT can preserve the benefits of attention while introducing computational advantages for parts of the network.

The compatibility results with GQA and the hybrid GQA-CAT-Alter accuracy snapshot are summarized in Appendix C.

## 5.3 Additional Baselines: Linear and Sparse Attentions

For completeness, we also experimented with a Linear Attention variant as in Katharopoulos et al. [2020], Choromanski et al. [2021], hoping to achieve $O(N)$ complexity under the same training conditions. However, on CLIP-L, we encountered repeated training instabilities (e.g., NaN loss values) and could not obtain stable convergence. Such issues are consistent with prior reports of kernel-based attention struggling to maintain numerical stability at larger scales.

We further examined representative sparse-attention methods such as BigBird Zaheer et al. [2020] and Longformer Beltagy et al. [2020]. However, these approaches introduce sequence-length–dependent sparsity hyperparameters (e.g., block size or global token selection), which complicate direct comparison under fixed training settings. As CAT is inherently hyperparameter-free by design, we focus on standard Attention to ensure a fair and controlled comparison.

Hence, we omit further comparisons with Linear and sparse attention variants in the main results.

## 5.4 Discussion

Taken together, these results show that **CAT** can reduce complexity while maintaining or even improving accuracy in settings with simpler token mixing requirements, particularly when using average pooling or masked inputs. Notably, *avg* pooling often outperforms *token* pooling across our baselines (Tab. 1), highlighting a broader trend in vision models that prefer global summarization over relying on a specialized classification token. Hence, the fact that CAT excels under *avg* pooling is especially significant: it aligns well with designs where tokens are mixed more globally, suggesting broad applicability in future ViT-like architectures. In addition, the strong gains in the *masked* setup

in language modeling are particularly noteworthy, given the recent surge of competitive Transformer variants that also adopt masked language modeling objectives Nie et al. [2025].

Meanwhile, **CAT-Alter** offers a balanced compromise, often exceeding standard attention's performance in both vision and language tasks. We hypothesize that employing CAT layers for parts of the network unlocks notable efficiency gains, while retaining the expressive power of full attention elsewhere. This hybrid-layer behavior aligns with the snapshots in Appendix C and mirrors findings in recent Transformer/SSM hybrids such as H3 Fu et al. [2023] and Jamba Lenz et al. [2025], where mixing attention with state-space components yields stronger or more stable performance than either mechanism alone. Achieving high accuracy under masked language modeling underscores CAT's potential for next-generation models where masking or simplified token interactions are central to performance.

# 6  Ablation Study

Thus far, we have primarily used a combined query-key projection $W_A$ alongside a separate value matrix $\mathbf{W_V}$. However, one can define or omit independent $\mathbf{Q}, \mathbf{K}$, and $\mathbf{V}$ in various ways, for example, by adopting an averaged key or partially splitting the query and key. To clarify how these design choices affect performance and parameter count, we conduct an ablation on ViT CLIP-L, comparing:

- **qkv (Averaged-Key)**: Retains separate query, key, and value modules (with $3D^2$ parameters), leveraging circular convolutions for sub-quadratic complexity. Its explicit q-k-v structure facilitates simpler integration into existing cross-attention pipelines compared to CAT's merged-query design.
- **qv (CAT)**: Our default approach (Sec. 4), where Q is absorbed into a single projection $W_A$, while V remains separate.
- **q only** or **v only**: Attempts for comparison to simplify by learning parameters either for the query or value, but relying on a dimension proportional to $N$. We denote these strategies as 'q-only' and 'v-only' for brevity.

For completeness, we summarize the formulations of each variant in the same notation as Sec. 4. Let $\mathbf{X} \in \mathbb{R}^{N \times D}$ denote the input sequence, let $\mathrm{circ}(\cdot)$ represent the circulant operator, and let $\mathrm{mean}(\cdot)$ represent the mean operator from $\mathbb{R}^{N \times D}$ to $\mathbb{R}^{1 \times D}$.

$$\text{(1) circular qkv (Averaged-Key):} \quad \mathbf{Q} = \mathbf{XW_Q}, \ \mathbf{K_{avg}} = \mathrm{mean}(\mathbf{XW_K}),$$
$$\mathbf{Z}^\star = \mathrm{softmax}\Big(\tfrac{\mathbf{QK_{avg}^\top}}{\sqrt{D_k}}\Big), \ \mathbf{F} = \mathrm{circ}(\mathbf{Z}^\star)\mathbf{XW_V};$$
$$\text{(2) circular qv (CAT):} \quad \mathbf{Z}^\star = \mathrm{softmax}(\mathbf{XW_A}), \ \mathbf{F} = \mathrm{circ}(\mathbf{Z}^\star)\,\mathbf{XW_V};$$
$$\text{(3) circular q only:} \quad \mathbf{Z}^\star = \mathrm{softmax}(\mathbf{XW_A}), \ \mathbf{F} = \mathrm{circ}(\mathbf{Z}^\star)\mathbf{V_T};$$
$$\text{(4) circular v only:} \quad \mathbf{Z}^\star = \mathrm{softmax}(\mathbf{Z_T}), \ \mathbf{F} = \mathrm{circ}(\mathbf{Z}^\star)\,\mathbf{XW_V},$$

where $\mathbf{V}_T \in \mathbb{R}^{N \times D}$ and $\mathbf{Z}_T \in \mathbb{R}^{N \times 1}$ are trainable parameters replacing $\mathbf{XW_V}$ and $\mathbf{XW_A}$.

All variants share the same circulant attention mechanism but differ in how query–key projections are parameterized. The qkv version retains full independence among $(\mathbf{W_Q}, \mathbf{W_K}, \mathbf{W_V})$; CAT (qv) merges query and key through $\mathbf{W_A}$; q-only and v-only restrict learnable parameters to a single branch.

**Observations.** While splitting into qkv yields a slight performance edge, it reintroduces learnable parameters. Conversely, q-only or v-only designs compromise accuracy and can inflate parameter requirements with respect to $N$. Therefore, our qv approach provides a practical balance, effectively emulating attention's capacity without incurring quadratic parameter overhead.

1. **qkv vs. qv:** Retaining fully distinct query, key, and value (qkv) yields the highest accuracy (0.696) but also the largest parameter budget ($3D^2$). In contrast, our qv design maintains nearly the same accuracy (0.694) with fewer specialized matrices and matrix operations.
2. **q or v alone:** Eliminating either query or value parameters reduces the complexity of one component but allocates learnable dimensions proportional to $N$. This approach not only

falls behind in accuracy (down to 0.637 or 0.625) but also scales poorly for larger sequence lengths.

3. **Balancing cost and performance:** The qv (CAT) configuration achieves a good trade-off, preserving global softmax weighting, keeping complexity at $O(N \log N)$, and attaining competitive accuracy.

**Parallels to Input-Based Control.** Beyond these core ablations, we note a conceptual parallel to *state-space models* such as Mamba Gu and Dao [2023, 2024], which incorporate input-driven control signals to update internal states more dynamically. Specifically, Mamba injects control variables from the input directly into its state equations, often improving accuracy via data-dependent updates. In a similar vein, our **qv** configuration merges query-like transformations ($\mathbf{Q}$) with the primary value projection ($\mathbf{V}$), effectively supplying an input-driven mechanism to modulate attention weights on a per-token basis. We hypothesize that this design fosters the same kind of dynamic control observed in Mamba, potentially explaining the *qv* variant's strong performance despite fewer learnable parameters than *qkv*. Exploring this connection, e.g., by systematically contrasting different input-injection strategies across attention and state-space frameworks, represents an intriguing direction for future research.

# 7 Conclusion

We introduced Engineering-Isomorphic Transformers (EITs), a framework that retains the essential softmax-based attention structure while reducing theoretical complexity below $O(N^2)$. Within this perspective (Sec. 2), our CAT results (Appendix B) indicate that a softmax-preserving, below-quadratic design can be both efficient and competitive at scale. As a concrete realization, our CAT leverages FFT-based circular convolutions to achieve $O(N \log N)$ runtime. Through extensive experiments on ImageNet-1k and WikiText-103, we demonstrated that CAT can match or surpass standard attention under simpler token operations (e.g., average pooling, masked language modeling), an important result, given that *avg* pooling increasingly outperforms token-based approaches in certain vision tasks, and that masked language modeling has become a highly competitive paradigm in recent Transformer research. Furthermore, our partial substitution scheme (CAT-Alter) remained robust across broader scenarios, at times exceeding full attention. An ablation study (Sec. 6) further revealed that merging query-key (qv) offers a practical balance of parameter efficiency and accuracy, outperforming alternatives like $k$-only or $v$-only projections. We believe this qv-based CAT design effectively emulates key aspects of standard attention without incurring quadratic complexity or parameter overhead.

**Open Challenges and Scalability.** Although CAT ensures sub-quadratic runtime in principle, validating its training stability and accuracy for extremely large sequences (e.g., tens of thousands of tokens) remains an open challenge. In particular, understanding how CAT scales in both efficiency and representational quality at these lengths would further solidify its real-world applicability Gu and Dao [2023]. Another promising direction is integrating CAT with other efficient attention variants, such as sparse, low-rank, or kernel-based methods, to push large Transformer architectures even further. From an implementation standpoint, hardware-optimized FFT kernels on GPUs or specialized accelerators could amplify the observed gains Dao et al. [2022]. Beyond language and vision, applying EITs principles to domains like speech or time-series forecasting may unveil new advantages of sub-quadratic models. We hope our work sparks further innovation in balancing efficiency and representational power, positioning *EITs* as a versatile foundation for the next generation of scalable deep learning.

**Broader Impact and Future Work.** While our sub-quadratic approach lowers the computational barrier and can democratize large-scale model training, it may also inadvertently accelerate the widespread usage of resource-intensive models, potentially increasing energy consumption and enabling misuse in certain applications. Addressing these concerns will require careful consideration of environmental sustainability, governance, and ethical guidelines in future work. We believe that ongoing efforts to develop hardware-optimized kernels and efficient deployment strategies will complement our method, helping to ensure that sub-quadratic Transformers can be harnessed responsibly and sustainably.

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

## A    Visualization of Attention Maps

We provide qualitative visualizations to illustrate how CAT and CAT-Alter differ from standard Self-Attention in terms of structural attention patterns. For CLIP-B (ViT-B/16), we visualize token-to-token attention maps at the patch level ($196{\times}196$). Each bottom panel in Figs. 3a–3b shows a $12{\times}12$ mosaic: rows correspond to attention heads ($h{=}1\ldots12$, left to right), and columns correspond to layers ($l{=}1\ldots12$, top to bottom). All maps are computed from per-head attention weights after softmax and are individually min–max normalized for visualization, using a shared colormap across methods. No averaging across heads or layers is applied before plotting.

In these visualizations, standard Self-Attention typically exhibits two canonical motifs: (1) woven, grid-like stripes representing cross-token interactions, and (2) near-identity diagonal bands capturing self-alignment. CAT, in contrast, produces more regular and shift-like diagonal structures that reflect its circulant formulation. While visually distinct, CAT achieves comparable or higher accuracy, as reported in the main tables. CAT-Alter combines both behaviors: it retains structured diagonal patterns in early layers while showing more globally diffuse attention in deeper layers. Interestingly, this mixture of localized and global dynamics coincides with its superior overall performance among all compared mechanisms, perhaps because combining fine-grained local structure with broader global context yields a more balanced representation.

## B    Runtime and Efficiency Analysis

To provide quantitative evidence of CAT's computational advantages, we present detailed runtime and memory profiles that clarify the extent of its speed and memory benefits over standard Self-Attention. All measurements were conducted on **NVIDIA V100 GPUs** (FP16 precision, batch size = 32) using the **AdamW** optimizer. Experiments used **PyTorch 2.7.0 + cu126 + cuDNN 9.5.1**, with **cuFFT** as the FFT backend for CAT and **FlashAttention** (torch.nn.functional.scaled_dot_product_attention) for Self-Attention baselines. We measure iteration time as the sum of the forward, backward, and optimizer passes, averaged over 10 steps. Table 4 reports this aggregate figure.

Table 4 summarizes runtime and memory usage for CLIP-L and GPT-2 small. CAT consistently reduces iteration time by 10–25% compared with Self-Attention, and the cuFFT implementation additionally lowers peak memory by about 25%.

Nonetheless, two practical bottlenecks still remain: (1) limited mixed-precision support, where performance degrades for sequence lengths not equal to powers of two. (2) Overhead for short sequences, where the gather version can outperform the FFT version.

Looking ahead, future GPU architectures with optimized FFT kernels and planning caches are expected to further accelerate the $O(N \log N)$ variant, especially for long-sequence applications.

## C    Compatibility with FlashAttention and GQA

Efficient attention mechanisms are often combined in practice to balance speed, memory, and accuracy. To facilitate adoption and to position CAT among widely used baselines, we note that CAT integrates with both FlashAttention and GQA (Grouped-Query Attention), and is expected to offer complementary efficiency gains when used together. Below we discuss how CAT interacts with each method, relate these observations to our FFT-based findings, and present a concise computational sketch together with small empirical results.

### C.1    FlashAttention Integration

FlashAttention's streaming formulation of exact softmax eliminates the need to materialize the full $N{\times}N$ attention map, substantially reducing memory traffic. A similar idea can be applied to CAT: implementing a *streaming circulant* variant allows the circular mixing to be computed on the fly without constructing intermediate matrices. This approach reduces transient buffers in the gather step and lowers peak memory, which is expected to provide *FlashAttention-like* benefits in runtime and memory efficiency, particularly when FFT planning overhead dominates at small sequence lengths (cf. Appendix B).

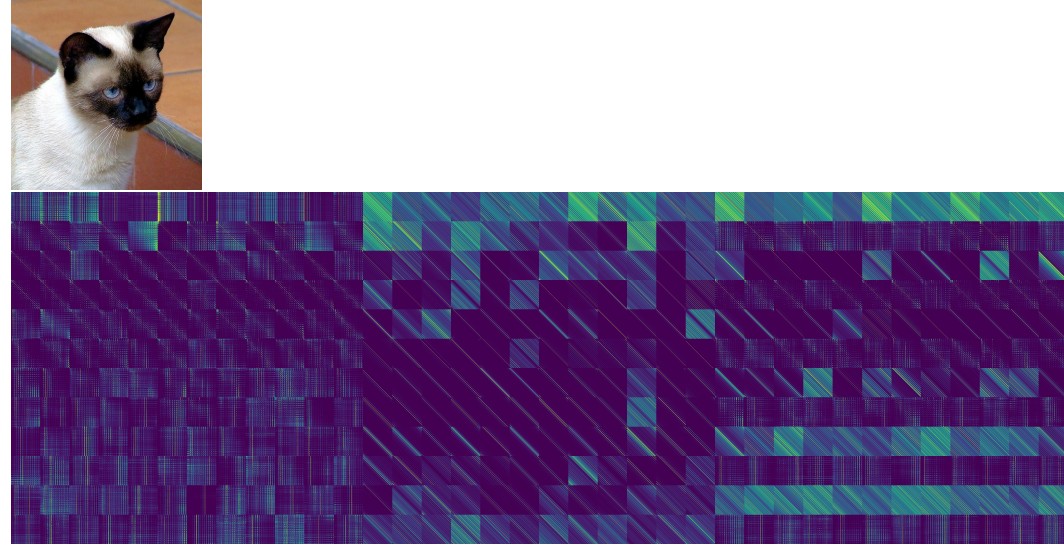

(a) **Attention-map visualization, input: siamese cat.**

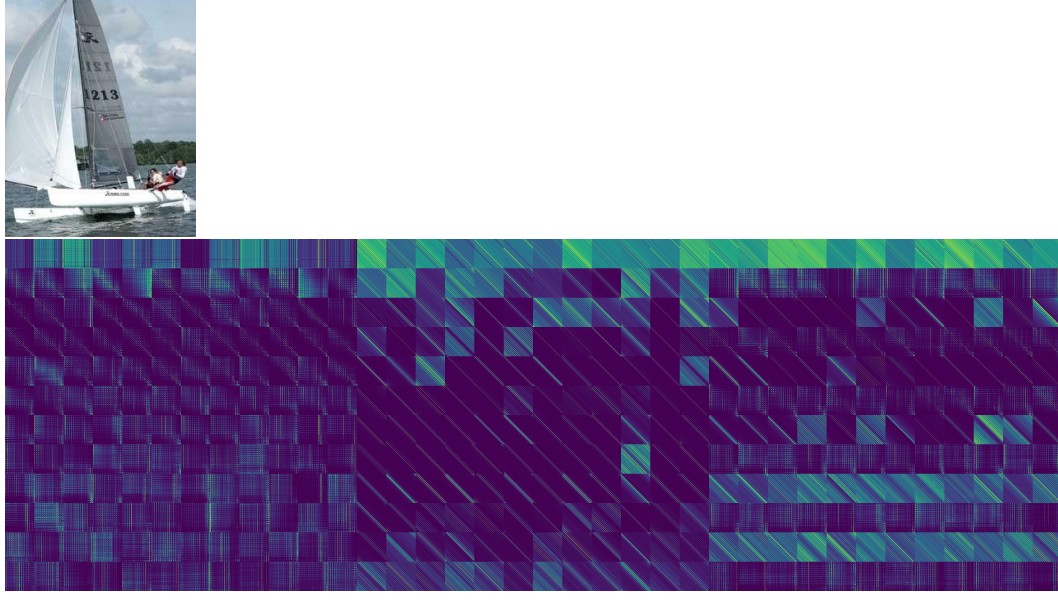

(b) **Attention-map visualization, input: catamaran.**

Figure 3: **Attention-map visualizations using CLIP-B (average pooling).** Top-left: input image. Bottom row (left→right): *Self-Attention*, *CAT*, and *CAT-Alter*. Each panel shows 196×196 token-to-token attention maps arranged as a 12×12 grid: rows correspond to attention heads ($h=1\ldots12$) within each layer, and columns correspond to layers ($l=1\ldots12$) from top to bottom. All maps share the same colormap and are normalized by min–max scaling.

| Variant | Param (M) | Peak Mem (MB) | Avg Time (ms) |
|---|---|---|---|
| *CLIP-L* | | | |
| Self-Attention (PyTorch MatMul) | 304.2 | 16654 | 1465.7 |
| Self-Attention (FlashAttention) | 304.2 | 13594 | 1467.5 |
| CAT (gather, $O(N^2)$) | **254.2** | 14931 | 1188.5 |
| CAT (cuFFT, $O(N \log N)$) | **254.2** | **11882** | **1187.9** |
| *GPT-2 small* | | | |
| Self-Attention (PyTorch MatMul) | 107.5 | 5289 | 521.6 |
| Self-Attention (FlashAttention) | 107.5 | 5259 | 531.0 |
| CAT (gather, $O(N^2)$) | **93.5** | **4651** | **441.8** |

Table 4: Runtime and memory profiles for **CLIP-L** and **GPT-2 small** on NVIDIA V100 GPUs (FP16, batch size = 32). CAT shows shorter iteration time than the Self-Attention baselines across both models, and the cuFFT variant further reduces peak memory on CLIP-L.

| Variant | Param (M) | Peak Mem (MB) | Avg Time (ms) |
|---|---|---|---|
| *CLIP-L* | | | |
| Self-Attention (PyTorch MatMul) | 304.2 | 16654 | 1465.72 |
| GQA ($K$=1/4) | 260.3 | 16510 | 1307.26 |
| GQA ($K$=1/16) | 257.0 | 16474 | 1270.39 |
| CAT (gather, $O(N^2)$) | **254.2** | **14931** | **1188.48** |

Table 5: Runtime snapshot for **CLIP-L** (V100 FP16, batch size = 32). CAT (gather) appears competitive and often faster than typical GQA configurations.

## C.2 GQA Enhancement (Runtime, Memory, Accuracy)

A compact projection-side compute model for the attention block can be written as

$$\text{GQA: } 2ND\left(D + 2DK\right), \qquad \text{CAT: } 2ND\left(D + H\right),$$

where $N$ is sequence length, $D$ hidden size, $H$ number of heads, and $K$ the key-reduction ratio in GQA. In regimes where $2DK > H$ (typical in large models), CAT is generally not slower in practice and tends to use fewer parameters within the attention block. To check this trend empirically, Table 5 reports a like-for-like comparison of GQA and CAT under identical settings.

The empirical results are consistent with the analytical model: CAT achieves lower runtime and comparable or smaller parameter counts than GQA, suggesting that their theoretical relation holds in practice.

In this small-scale sweep, **GQA-CAT-Alter** (i.e., replacing the attention component of CAT-Alter with GQA) achieves both reduced parameter count and improved runtime compared with standard GQA, while maintaining comparable accuracy. Although limited in scale, this consistent trend indicates that incorporating CAT principles into GQA is a promising direction for further exploration.

## C.3 Summary

Taken together, these observations suggest that replacing standard GQA with CAT-based designs (GQA-CAT-Alter) offers consistent advantages in runtime, memory, and parameter efficiency while maintaining comparable or better accuracy. This direction appears to be a viable and promising path for extending GQA-style models, while potentially enabling FlashAttention-like streaming benefits and helping mitigate FFT-related memory overheads for small sequences.

## D Permutation Equivariance and Domain Applicability

When we reorder the input tokens, should the model's output stay unchanged, follow the same reordering, or respond differently? This choice fixes the model's inductive bias toward order: it specifies whether the representation should ignore the order (invariance), respect it up to relabeling (equivariance), or treat different orders as different meanings. Match the symmetry to the task: CNNs

| Model | Pool Type | Mechanism | Setting | Acc. |
|-------|-----------|-----------|---------|------|
| CLIP-B | token | Attention | – | *0.574* |
| CLIP-B | token | CAT-Alter | – | *0.582* |
| CLIP-B | token | GQA | $K{=}1/4$ | 0.561 |
| CLIP-B | token | GQA-CAT-Alter | $K{=}1/4$ | **0.571** |
| CLIP-B | avg | Attention | – | *0.638* |
| CLIP-B | avg | CAT-Alter | – | *0.662* |
| CLIP-B | avg | GQA | $K{=}1/4$ | 0.629 |
| CLIP-B | avg | GQA-CAT-Alter | $K{=}1/4$ | **0.658** |

Table 6: Accuracy comparison on **CLIP-B** (token and average pooling). CAT-Alter consistently outperforms vanilla MHA, while the hybrid **GQA-CAT-Alter** improves upon standard GQA in both pooling settings. These small-scale results are consistent with our runtime findings, indicating that CAT-based extensions of GQA are empirically stable and practically efficient.

are translation-equivariant at the layer level and can yield translation invariance after pooling/readout; position-free Transformers for permutation-equivariant, order-agnostic inputs (e.g., sets, point clouds). In natural language, by contrast, the order of words often carries meaning (e.g., "dog bites man" vs. "man bites dog"; the scope of items like *only* or negation; long-range dependencies), so the full permutation symmetry must be **broken** by injecting positional information (absolute or relative). This makes relative order and distance representable and typically yields small but consistent gains in practice. Spelling out the required symmetry up front clarifies what is gained or lost when we modify the attention mechanism or add positional encodings.

Without positional signals, a self-attention layer is equivariant to any permutation in the full symmetric group $S_N$, and becomes permutation invariant only when paired with an invariant read-out (e.g., global pooling). By contrast, CAT's circulant mixing makes the layer equivariant to cyclic shifts in $C_N$; with an order-agnostic read-out the overall mapping becomes invariant to cyclic shifts. Since $C_N$ is a strict subgroup of $S_N$, cyclic-shift equivariance/invariance does **not** imply equivariance/invariance to arbitrary permutations.

When permutation symmetry is required, for true set-structured inputs or certain graph/particle systems, the symmetry must be engineered into the model explicitly. Two lightweight, design-intent examples are: (i) imposing a canonical order via differentiable sorting (e.g., SoftSort) before applying CAT; and (ii) adopting a hybrid pipeline in which an order-assigning encoder precedes CAT and a permutation-invariant read-out aggregates the final representation.

