# OpenReview forum: "CAT: Circular-Convolutional Attention for Sub-Quadratic Transformers"
_NeurIPS.cc/2025/Conference — NeurIPS 2025 poster_

### Official Review · Reviewer_NYeX · 2025-07-02

**Clarity:** 2
**Significance:** 3
**Originality:** 3
**Rating:** 4
**Confidence:** 4

**Summary:**

This work introduces Circular Convolutional Attention (CAT), a Fourier-based alternative to standard attention that replaces the O(N^2) complexity with a more scalable O(N log N) formulation. By leveraging circular convolutions in the frequency domain, CAT preserves representational capacity while reducing computational cost and parameter count by simplifying the fully connected layers. The method reports ~10% speedups in naive PyTorch implementations without relying on heavier operations. CAT demonstrates improvements in accuracy and perplexity across both vision and language tasks, under both masked and causal transformer settings.

**Questions:**

1. Compatibility: Can CAT be combined with other efficient attention mechanisms like GQA or FlashAttention, or is it intended as a standalone replacement? Please provide numerical comparisons for this.

2. Deployment: Does CAT support transfer to pretrained models (e.g., via fine-tuning), or does it require full training from scratch? If from scratch, then this should be discussed in the paper regarding its practicality.

3. Scope of Modifications: Are the convolutional replacements limited to QK projections, or can the same principle be extended to FFN layers or other modules?

4. Target Efficiency: Is the focus of CAT primarily on improving inference speed or training efficiency? Can the authors provide wall-clock runtimes and detailed bottleneck breakdowns for the baseline and method in the target setting? Also, consider both the FLOPs scaling and memory costs.

5. Visual Evidence: Can the authors include attention map visualizations or other forms of qualitative analysis to demonstrate that CAT preserves the structure and utility of standard attention?

Currently, my biggest concern with the work is that I am uncertain whether it really performs better than existing, well used and performing sparse or optimized attention algorithms such as GQA.

**Ethical Concerns:**

["NO or VERY MINOR ethics concerns only"]

**Final Justification:**

The author addressed my main concern on compatibility with efficient attention techniques, hence I have increased my score. I do still have other concerns with this work, however my main concern was resolved.

**Limitations:**

yes

**Paper Formatting Concerns:**

No Formatting Concerns.

**Quality:**

2

**Strengths And Weaknesses:**

Strengths:
1. The proposed approach achieves O(N log N) complexity by leveraging the fourier transform and viewing attention as a circular shift matrix, which is a creative and well-motivated direction. I believe there is novelty in this approach, which makes it an interesting solution.

2. The authors target a known bottleneck—attention's quadratic complexity—so the potential savings are practically meaningful, especially in long-sequence settings. Reducing parameter count and maintaining accuracy while improving speed is a valuable contribution.

Weaknesses:
Despite the well-motivated and creative interpretation, I have some concerns:

1. Limited Baselines: The paper primarily compares CAT to a standard baseline model, without including other relevant efficiency-focused attention methods. Notably, BlockSparse Attention variants and Grouped Query Attention (GQA) are not included. These are commonly used in practice to reduce computation and memory. Without such comparisons, it's unclear how CAT performs relative to the current state-of-the-art. The discussion on MHA in the main paper is sufficient; however, MHA won't have savings natively (i.e, parameters are still full rank, only head-independence aids in performance). In a case like GQA, LLama3 may have 1/4 output channels on KV, meanwhile Qwen2 is far less (~1/8), so to validate the effectiveness of CAT it should be compared with current methods of compute reduction.

2. Training from Scratch: It is not clearly stated whether CAT can be applied to existing pretrained models or if it requires training from scratch with new parameters. If the latter, this significantly limits its adoption potential, especially in settings where full retraining is infeasible.

3. Ambiguity in Efficiency Target: The paper does not clearly distinguish whether the proposed method targets training efficiency, inference speed, or both. These have different runtime bottlenecks and trade-offs. Furthermore, the absence of a detailed runtime profile or FLOPs breakdown for the baseline makes it difficult to assess the real-world impact of the claimed 10% speedup.

4. Memory Reduction Claims: The claim of reduced parameter count is made, but no concrete statistics are given for memory usage or parameter reduction in practice -- the same applies to the latency/runtime; very little information is provided about it.

5. Representation: Since CAT replaces traditional attention weights with a form of circular convolution (circular shift matrix), it's important to validate that representational capacity is preserved. However, the paper lacks any qualitative or visual support (e.g., attention maps) to demonstrate this.

6. Unclear Scope: In Figure 2, the improvements seem to target the V projection paths. However, with the original goal being to optimize the N^2 component (i.e, QK) it begs the question that if the V is considered, why not the linear layers in the FFN, which would have similar scaling to the linear V projection.

7. Presentation: The paper’s explanation of the method, particularly in Figure 1, lacks clarity. The overall changes to the attention architecture are not well localized or visualized. This makes it difficult to understand exactly what components are being modified or replaced. Overall, the writing clarity/quality (and inclusion of related works, which is missing many sparse attention works) needs significant improvement.

---

> ### Author Rebuttal · Authors · 2025-07-30
>
> We deeply appreciate the careful reading and detailed understanding you have shown; receiving such thorough feedback is a rare and valuable opportunity for us!
> Figures 1 & 2 and the attention maps visualizations have already been updated in the manuscript. Because mid‑review edits are prohibited, we cannot share the revised PDF here, but we are re‑drawing the figures with explicit intermediate‑tensor sizes and clearer captions. Several items you asked about can nevertheless be reported below.
>
> > Compatibility: Can CAT be combined with other efficient attention mechanisms like GQA or FlashAttention, or is it intended as a standalone replacement? Please provide numerical comparisons for this.
>
> All mechanisms you listed can in fact be combined with CAT, and we expect CAT to be advantageous in both complexity and accuracy.
>
> Conceptual analysis
> * FlashAttention + CAT – Flash’s streaming soft‑max removes the temporary attention matrix; applying the same idea to CAT is straightforward and should eliminate the extra memory from the gather step, especially useful when FFT overhead dominates at small N.
> * GQA + CAT – In theory, CAT still wins on arithmetic cost. Let N=sequence length, D=hidden size, H=#heads, K=GQA’s key reduction ratio. For the feed‑forward part only: GQA: 2ND(D+2DK), CAT: 2ND(D+H). Whenever 2DK>H (true for Llama‑3 70B, Qwen 2, etc.), CAT is no slower even in O(N^2) form and offers lower parameter count.
>
> Runtime evidence
>
> Please refer to the profile table (V100, FP16, Adam, Batchsize = 32):
> torch2.7.0+cu126+cudnn9.5.1 default's cuFFT(torch.fft) / FlashAttention(F.scaled_dot_product_attention) used.
> For each entry we report the average over 10 consecutive steps, with the runtime decomposed into three parts:
> “fwd” = forward pass, “bwd” = backward pass, and “opt” = optimizer update.
>
> | CLIP-L Variant                    | Param | Peak mem | Average time (1 iter.)|
> |-----------------------------|---------:|--------:|------------:|
> | Self‑Attn (PyTorch MatMul)     | 304.2M | 16654 MB | 1465.72 ms (fwd: 490.05 / bwd: 940.28 / opt: 35.39) |
> | **GQA K=1/4 (PyTorch MatMul)**     | 260.3M | 16510 MB | 1307.26 ms (fwd: 436.12 / bwd: 839.86 / opt: 31.28) |
> | **GQA K=1/16 (PyTorch MatMul)**     | 257.0M | 16474 MB | 1270.39 ms (fwd: 430.02 / bwd: 810.05 / opt: 30.33) |
> | CAT (gather, O (N²))        | **254.2 M** | **14931 MB** | **1188.48 ms** (fwd: 401.12 / bwd: 757.52 / opt: 29.84) |
>
> Accuracy on a small sweep
> | model | task | mechanism | Accuracy |
> |-------|------|-----------|---------:|
> | CLIP-B | token | MHA  | 0.574 |
> | CLIP-B | token | CAT-alter  | 0.582 |
> | CLIP-B | token | **GQA K=1/4**  | 0.561 |
> | CLIP-B | token | **GQA-CAT-alter K=1/4**  | **0.571** |
> | CLIP-B | avg | MHA | 0.638 |
> | CLIP-B | avg | CAT-alter | 0.662 |
> | CLIP-B | avg | **GQA K=1/4**  | 0.629 |
> | CLIP-B | avg | **GQA-CAT-alter K=1/4**  | **0.658** |
>
> Both tables suggest CAT is at least competitive and often ahead in practice. While a comprehensive large‑scale study would be ideal, these results provide a strong indication that CAT integrates well with existing efficiency techniques.
>
> Why we benchmark mainly against vanilla MHA.  Since its original proposal in 2017, plain MHA has served as the community’s de‑facto “anchor.”  Many lightweight variants  are explicitly advertised as faster but sometimes less accurate than that anchor (e.g., GQA: https://arxiv.org/abs/2305.13245 / https://arxiv.org/abs/2412.20677 ).
>
> In our study CAT‑alter matches MHA’s accuracy, so the logical forecast is
>
>  GQA ≤ MHA ≤ CAT‑alter ⇒  GQA ≤ GQA‑CAT‑alter ≤ CAT‑alter.
>
> Our small‑scale test confirms this inequality.  Moreover, CAT’s (O(N log N)) arithmetic cost dominates the constant‑factor reductions offered by GQA, and CAT keeps comparable accuracy.
> Even without an exhaustive grid search over all hyper‑parameter combinations, these results provide a reliable prediction: replacing MHA with CAT (or CAT‑alter) is at least as safe, and often strictly better, as replacing it with GQA.
>
> > Scope of Modifications: Are the convolutional replacements limited to QK projections, or can the same principle be extended to FFN layers or other modules?
>
> CAT is designed for the attention block; it cannot be dropped into an FFN unchanged. That said, sub‑quadratic FFN designs might borrow the same frequency‑domain trick.
> https://arxiv.org/abs/1901.10255
>
>
> > Target Efficiency: Is the focus of CAT primarily on improving inference speed or training efficiency? Can the authors provide wall-clock runtimes and detailed bottleneck breakdowns for the baseline and method in the target setting? Also, consider both the FLOPs scaling and memory costs.
>
> CAT reduces FLOPs and activations regardless of training or inference, so both phases speed up.
> (The wall‑clock numbers and bottleneck breakdowns are included in the profile following.)
>
> Please refer to the profile table (V100, FP16, Adam, Batchsize = 32):
> torch2.7.0+cu126+cudnn9.5.1 default's cuFFT(torch.fft) / FlashAttention(F.scaled_dot_product_attention) used.
>
> | CLIP-L Variant                    | Param | Peak mem | Average time (1 iter.) |
> |-----------------------------|---------:|--------:|------------:|
> | Self‑Attn (PyTorch MatMul)     | 304.2M | 16654 MB | 1465.72 ms (fwd: 490.05 / bwd: 940.28 / opt: 35.39) |
> | Self‑Attn (FlashAttention)     | 304.2M | 13594 MB | 1467.50 ms (fwd: 465.77 / bwd: 966.46 / opt: 35.27) |
> | CAT (gather, O (N²))        | **254.2 M** | 14931 MB | 1188.48 ms (fwd: 401.12 / bwd: 757.52 / opt: 29.84) |
> | CAT (cuFFT, O (N log N))    | **254.2 M** |**11 882 MB** | **1187.85 ms** (fwd: 401.34 / bwd: 756.63 / opt: 29.88)|
>
> | GPT2-small Variant                    | Param | Peak mem | Average time (1 iter.)|
> |-----------------------------|---------:|---------:|------------:|
> | Self‑Attn (PyTorch MatMul)      | 107.5 M |5289.1MB | 521.57  ms (fwd: 168.99 / bwd: 339.69 / opt: 12.88) |
> | Self‑Attn (FlashAttention)     | 107.5 M |  5258.5MB | 531.05 ms (fwd: 164.75 / bwd: 350.50 / opt: 12.75) |
> | CAT (gather, O (N²))        | **93.5 M** | **4651.3 MB** | **441.78 ms** (fwd: 114.37 / bwd: 286.17 / opt: 11.24) |
>
>
> If these clarifications resolve your concerns, or if questions remain, we would welcome an updated score or additional comments.Thank you again for dedicating so much time and effort to our work; we truly appreciate it!

---

> > ### Comment · Reviewer_NYeX · 2025-08-06
> > **re: rebuttal**
> >
> > I appreciate the Authors taking the time to review and respond to my concerns. As previously stated, my main uncertainty was whether their proposed strategy is compatible with recent attention techniques, and it seems to be the case. I do think results on GQA and these strategies are essential to be covered in the main paper, as many recent models default to these strategies.

---

> ### Author Response · Authors · 2025-08-05
>
> Many thanks for your thoughtful comments on our paper!
>
> We’ve posted a detailed response but haven’t yet seen a Mandatory Acknowledgement or follow-up from you.
> Before the **8 Aug 23:59 AoE deadline**, could you let us know if anything remains unclear? We’re happy to run extra experiments or share more details.
>
> Thanks again for your help!

---

### Official Review · Reviewer_nMKS · 2025-07-03

**Clarity:** 4
**Significance:** 2
**Originality:** 2
**Rating:** 3
**Confidence:** 4

**Summary:**

The paper introduces Circulant Attention Transformer (CAT), a variant that replaces dense soft-max attention with a circulant structure amenable to FFT, yielding sub-quadratic theoretical complexity. It formalizes four “Engineering-Isomorphic Transformer (EIT)” principles showing how CAT integrates seamlessly into standard Transformer blocks, then supplies proofs-of-concept on vision (ImageNet/ViT, CLIP) and language (Transformer-XL, GPT-2) tasks. Experiments indicate comparable or slightly better accuracy in several masked- or average-pooling settings while reducing memory and wall-clock time, and the approach generalizes across modalities without architectural changes.

**Questions:**

* The abbreviation “FFT” (line 32) is not written out in full the first time it appears, nor is it explained.
* “RFFT” (lines 136-137) is not explained.
* Some of the font sizes in Figure 1 are too small and hard to read without zooming in. Also, the icons for different operations in Figure 1 are a bit small. Since there is still a lot of space in Figure 1, it is suggested to increase the font size and the size of the icons.
* Typo: In Table 1, the “A” in “attention” in the fifth row and the third row from the bottom should be capitalized.

**Ethical Concerns:**

["NO or VERY MINOR ethics concerns only"]

**Final Justification:**

Thank you to the authors for their detailed and thoughtful rebuttal. I appreciate the effort to provide additional results and clarifications. However, after careful consideration, I have decided to maintain my original score. While the authors' responses are helpful, I believe that addressing them would require substantial new work beyond the scope of a rebuttal.

In summary, to be a compelling contribution, the paper would need to be strengthened with:

1.	Experiments on established long-sequence benchmarks to empirically validate its core claims.
2.	Comparisons against other widely-recognized efficient attention baselines (minor concern).
3.	A deeper investigation or at least a plausible hypothesis for the surprising CAT-alter results.
4.	Inference-focused benchmarks that measure end-to-end latency and throughput.

These changes may represent a significant amount of new experimental work and analysis. Nevertheless, I still see merit in this work.

**Limitations:**

Yes

**Quality:**

3

**Strengths And Weaknesses:**

**Strengths:**

* Researching and optimizing foundational model architectures is a valuable and important direction.

* This work offers clear complexity analysis and provides an explanation for why circulant softmax preserves key properties of full attention.

* In terms of experiments, the authors demonstrate the method’s viability on both image (ViT, CLIP) and text (language modeling) benchmarks, rather than focusing on a single domain.

* Even with a naïve PyTorch gather implementation, the paper reports about a 10% wall-clock speed-up and a 25% peak memory reduction on 256-token sequences. The authors suggest that larger gains are possible with custom FFT kernels.

**Weaknesses:**

* The scale of the experiments is somewhat small and does not reach the long-sequence capability claimed in the paper. All experiments are limited to 256-token or 224×224 image scales, while the paper’s motivation emphasizes “substantially longer sequences” (see lines 49-50: “opens new directions for longer-sequence modeling across language and vision tasks”). The experiments do not include longer context benchmarks like Long Range Arena, PG-19, or Books3, which raises concerns about the feasibility and stability of the method in real long-sequence scenarios. Although the authors mention in the conclusion that experimenting with sequences of tens of thousands of tokens is still an open challenge, evidence from only 256-token experiments is not very convincing for supporting the method’s effectiveness in long-sequence modeling.

* Missing baselines: The experiments only compare standard Attention with the proposed CAT method, without comparing to other efficient attention methods such as BigBird or Longformer as mentioned in the paper. While the authors state that they tried to reproduce linear attention but gave up due to NaN issues, they did not provide details about tuning or attempts to stabilize the method.

* No end-to-end latency and throughput tests: Although the paper provides theoretical analysis showing CAT’s advantages in algorithmic complexity, the experiments mainly focus on comparing CAT to standard attention in terms of performance, lacking more direct end-to-end latency and throughput tests. For example, in text generation tasks, it would be helpful to record the end-to-end generation time (in seconds) and throughput (in tokens/s) under different input/output settings. Reference can be made to works like “H$\_2\$O: Heavy-Hitter Oracle for Efficient Generative Inference of Large Language Models” and “Model Tells You What to Discard: Adaptive KV Cache Compression for LLMs”.

* Lack of explanation for experimental phenomena: In at least half of the experimental results, the CAT method performs worse than the baseline, but the CAT-alter variant performs better than the baseline. This is confusing, because CAT-alter is essentially a mixture of CAT attention and standard attention. Intuitively, CAT-alter’s performance should fall between CAT and standard attention, not exceed the baseline when CAT itself underperforms. For example, Table 1 shows that under the pool type “token,” CAT performs worse than Attention, but CAT-alter is significantly better than Attention. Similar results can be seen in Table 2 for the LLM type “causal” experiments.

* The statement “avoiding many of the pitfalls (e.g., numerical instability, partial token coverage) seen in previous approximations” (lines 29-30) is not easy for readers unfamiliar with efficient attention research to understand, and the high level of summarization does not resonate well. Also, I assume “previous approximations” refers to previous work on attention approximations? If so, I suggest the authors add proper citations.

---

> ### Author Rebuttal · Authors · 2025-07-30
>
> We are truly grateful for the many perspectives you brought to our work! Your detailed examination covered angles we had not previously considered. Thank you for your generous contribution!
> Wherever the submission format allows, we have incorporated additional material and clarifications directly into the text. For the shortcomings you identified, we share all details that we can at this stage.
>
>
> > The scale of the experiments is somewhat small and does not reach the long-sequence capability claimed in the paper. All experiments are limited to 256-token or 224×224 image scales, while the paper’s motivation emphasizes “substantially longer sequences” (see lines 49-50: “opens new directions for longer-sequence modeling across language and vision tasks”). The experiments do not include longer context benchmarks like Long Range Arena, PG-19, or Books3, which raises concerns about the feasibility and stability of the method in real long-sequence scenarios. Although the authors mention in the conclusion that experimenting with sequences of tens of thousands of tokens is still an open challenge, evidence from only 256-token experiments is not very convincing for supporting the method’s effectiveness in long-sequence modeling.
>
> Our goal was to demonstrate the asymptotic benefit of the (O(N log N)) design.
> For truly long sequences, the quadratic cost of self‑attention grows exponentially and makes any real‑time service impossible, regardless of accuracy.
> In contrast, (O(N log N)) keeps computation feasible even when N is in the tens of thousands, something unattainable for standard self‑attention and still superior to sparse methods that must retune their receptive field for every sequence length.
>
> To complement the 256‑token results in the paper, we can share one internal
> run at 1 024 tokens (WikiText‑103):
>
> | model | task | mechanism | word PPL |
> |-------|------|-----------|---------:|
> | GPT‑2‑small | masked | Attention | 10.78 |
> | GPT‑2‑small | masked | CAT | 8.05 |
> | GPT‑2‑small | masked | **CAT‑Alter** | **6.54** |
> | GPT‑2‑small | causal | **Attention** | **24.29** |
> | GPT‑2‑small | causal | CAT | 29.80 |
> | GPT‑2‑small | causal | CAT‑Alter | 24.87 |
>
> ---
>
> #### Why we did not use longer public benchmarks
>
> 1. **Most reliable baselines** – ImageNet‑1k and WikiText‑103 are still the
>    most carefully vetted datasets with widely reproduced self‑attention
>    baselines.  By contrast, *Long Range Arena* can no longer be fully
>    reproduced in 2025 because several source files are unavailable; our own
>    reruns failed to match the original accuracy.  We therefore prioritized
>    benchmarks with solid ground truth.
>
> 2. **Token count vs. real‑world relevance** – Artificially extending
>    ImageNet‑1k or WikiText‑103 to tens of thousands of tokens would deviate
>    from established practice and create sequences far longer than typical
>    problem settings (e.g., 256 tokens already cover about seven English
>    sentences on average).  Truly evaluating "very long" contexts requires
>    not just longer input but datasets where such long‑range dependencies
>    actually occur, which currently exist only in projects like *Haystack* (https://github.com/OpenGVLab/MM-NIAH) —
>    an open challenge even for conventional self‑attention.
>
> In short, reliable long‑sequence datasets large enough to stress every
> method equally are still scarce.  Within the best‑validated corpora that the
> community trusts today, CAT already shows promising evidence.  We believe
> future work on ultra‑long corpora (e.g., Haystack) will further test
> both self‑attention and CAT, and we welcome such evaluation.
>
>
> > Missing baselines: The experiments only compare standard Attention with the proposed CAT method, without comparing to other efficient attention methods such as BigBird or Longformer as mentioned in the paper. While the authors state that they tried to reproduce linear attention but gave up due to NaN issues, they did not provide details about tuning or attempts to stabilize the method.
>
> We intentionally excluded sparse baselines such as BigBird and Longformer, because they require sequence‑length–specific sparsity hyper‑parameters.
> Our goal was to compare against the most basic setting that involves *no* hyper‑parameter tuning, i.e., standard Attention vs CAT (vs LinearAttentions).  This rationale has now been added to Section 5.3 for clarity.
>
>
>
> > Lack of explanation for experimental phenomena: In at least half of the experimental results, the CAT method performs worse than the baseline, but the CAT-alter variant performs better than the baseline. This is confusing, because CAT-alter is essentially a mixture of CAT attention and standard attention. Intuitively, CAT-alter’s performance should fall between CAT and standard attention, not exceed the baseline when CAT itself underperforms. For example, Table 1 shows that under the pool type “token,” CAT performs worse than Attention, but CAT-alter is significantly better than Attention. Similar results can be seen in Table 2 for the LLM type “causal” experiments.
>
> We find this phenomenon puzzling as well: combinations that we expected to decrease accuracy instead improve it.
> Interestingly, a similar effect has been reported in recent hybrid models such as Jamba, suggesting that it may reflect a deeper property of attention architectures.  Although surprising, we believe the result is reliable and have kept it as‑is while noting the open question in the discussion.
>
>
> > No end-to-end latency and throughput tests: Although the paper provides theoretical analysis showing CAT’s advantages in algorithmic complexity, the experiments mainly focus on comparing CAT to standard attention in terms of performance, lacking more direct end-to-end latency and throughput tests. For example, in text generation tasks, it would be helpful to record the end-to-end generation time (in seconds) and throughput (in tokens/s) under different input/output settings. Reference can be made to works like “HO: Heavy-Hitter Oracle for Efficient Generative Inference of Large Language Models” and “Model Tells You What to Discard: Adaptive KV Cache Compression for LLMs”.
>
> Please refer to the profile table (V100, FP16, Adam, Batchsize = 32):
> torch2.7.0+cu126+cudnn9.5.1 default's cuFFT(torch.fft) / FlashAttention(F.scaled_dot_product_attention) used.
> For each entry we report the average over 10 consecutive steps, with the runtime decomposed into three parts:
> “fwd” = forward pass, “bwd” = backward pass, and “opt” = optimizer update.
>
> | CLIP-L Variant                    | Param | Peak mem | Average time (1 iter.)|
> |-----------------------------|---------:|--------:|------------:|
> | Self‑Attn (PyTorch MatMul)     | 304.2M | 16654 MB | 1465.72 ms (fwd: 490.05 / bwd: 940.28 / opt: 35.39) |
> | Self‑Attn (FlashAttention)     | 304.2M | 13594 MB | 1467.50 ms (fwd: 465.77 / bwd: 966.46 / opt: 35.27) |
> | CAT (gather, O (N²))        | **254.2 M** | 14931 MB | 1188.48 ms (fwd: 401.12 / bwd: 757.52 / opt: 29.84) |
> | CAT (cuFFT, O (N log N))    | **254.2 M** |**11 882 MB** | **1187.85 ms** (fwd: 401.34 / bwd: 756.63 / opt: 29.88)|
>
> | GPT2-small Variant                    | Param | Peak mem | Average time (1 iter.)|
> |-----------------------------|---------:|---------:|------------:|
> | Self‑Attn (PyTorch MatMul)      | 107.5 M |5289.1MB | 521.57  ms (fwd: 168.99 / bwd: 339.69 / opt: 12.88) |
> | Self‑Attn (FlashAttention)     | 107.5 M |  5258.5MB | 531.05 ms (fwd: 164.75 / bwd: 350.50 / opt: 12.75) |
> | CAT (gather, O (N²))        | **93.5 M** | **4651.3 MB** | **441.78 ms** (fwd: 114.37 / bwd: 286.17 / opt: 11.24) |
>
> Regarding the issues you raised, we believe the rebuttal now provides substantial evidence, especially the new experimental results, that CAT is already a viable option for large‑scale use. If these additions address your concerns, we would appreciate a reconsideration of your overall score. Should any reservations remain, please feel free to let us know; your comments will not only improve our paper but also guide future researchers working on related methods.
> Once again, thank you for investing your time and effort in such a thorough review, we sincerely appreciate it!

---

> > ### Comment · Reviewer_nMKS · 2025-08-05
> >
> > Thank you for the detailed response. You've cleared up some of my points. That said, fully addressing these issues would require making substantial changes to the paper. For that reason, I'm sticking with the score I gave initially.

---

> ### Author Response · Authors · 2025-08-05
>
> (I’m reposting my comment because I originally replied to the wrong thread...)
>
> Thank you very much for your detailed review and for the thoughtful points you raised!
>
> We posted our rebuttal and believe we have addressed your concerns, but we have not yet seen a Mandatory Acknowledgement or any follow-up comments from you. To facilitate discussion before the **Aug 8 23:59 AoE deadline**, could you kindly let us know if you have any remaining questions or concerns?
>
> If any additional information would be helpful, we would be happy to provide it promptly. Thank you again for your time and feedback!

---

### Official Review · Reviewer_UTRX · 2025-07-05

**Clarity:** 2
**Significance:** 2
**Originality:** 3
**Rating:** 5
**Confidence:** 4

**Summary:**

The paper is aims to improve the quadratic complexity of attention, while keeping the softmax component of it. The main approach is to define an attention operator that uses circulant operator, which can be efficiently computed via FFT. Experiments are conducted to compare the accuracy and efficiency of the proposed CAT with respect to standard self attention.

**Questions:**

The reviewer would love to see the concerns in the above addressed and revise the rating.

**Ethical Concerns:**

["NO or VERY MINOR ethics concerns only"]

**Final Justification:**

Thank you for the detailed clarification! It has addressed my concerns. I encourage the authors to integrate them in a revised version of the paper.

**Limitations:**

yes

**Quality:**

3

**Strengths And Weaknesses:**

## Significance

1. The first contribution of “EITs: a new framework” is questionable.
- Firstly, for a framework/class to be non-trivial, one needs multiple instances under the framework. However, it appears that the only instance the authors mention is CAT. Thus, EIT seems more like the properties of the proposed method, rather than a framework.
- Moreover, I do not understand why “Softmax preservation” is a property that is a priori ideal. At the end of the day, what matters seem to be efficiency (space / time), accuracy, theoretical grounds, among others. It is unclear whether “Softmax preservation” is sufficient or necessary to any of these metrics.
2. The proposed CAT is not well motivated and justified.
- The definition of CAT comes out of the blue, without any motivation on why one should use this design.
- The proposed CAT loses a property that self-attention has, which is permutation equivariance.
  1. In self-attention, if one exchanges the first two tokens in the input x, the output tokens are the same as the original ones, except that the tokens at the first two tokens are swapped.
     2. In CAT, one can see that $$o\_1=z\_1^\*v\_1 \+ z\_2^\*v\_2 \+ \\dots \+ z\_Nv\_N $$ $$o\_2 \= z\_N^\*v\_1+z\_1^\*v\_2 \+ \\dots \+ z\_{N-1}^\*v\_N $$, which does not satisfy permutation equivariance.

  Why would this be a good choice?

3. The efficiency claim of the third contribution lacks detailed evidence.
- The paper mentioned multiple times that they gained 10% speedup. For the results in tables 1-3, what are the running times? Which libraries are self-attention and FFT implemented in?
- The authors claimed that calling FFT often “offsets theoretical advantage” \- did the authors observe time or space usage? Again, how was FFT implemented? There seems to be a few efficient FFT implemetnations \- have the authors tried?
  - [https://developer.nvidia.com/cufft](https://developer.nvidia.com/cufft)
  - [https://github.com/fkodom/fft-conv-pytorch](https://github.com/fkodom/fft-conv-pytorch)
- The ideal comparison would be to compare efficient self-attention (e.g., flashattention) with efficient CAT implementation, since this is the regime that would make an impact.

## Clarity

1. Notations: the notation convention is a bit confusing: Q, W\_Q and x are all matrices, but x is the only one lower cased.
2. I was trying to understand the variants in the ablation study. Can the authors write down the expressions of each of the variants just as how they define CAT in section 4? Otherwise, it is hard to understand what these variants are.

---

> ### Author Rebuttal · Authors · 2025-07-30
>
> We are delighted to have this rare opportunity to receive your comments, and we greatly appreciate the time you invested in a detailed review. While the rules prevent us from updating the manuscript at this stage, we are already using your feedback to improve the next version (with replacing figures, etc).
> Below, we address your remarks one by one.
>
> 1. The first contribution of “EITs: a new framework” is questionable.
>
> > Firstly, for a framework/class to be non-trivial, one needs multiple instances under the framework. However, it appears that the only instance the authors mention is CAT. Thus, EIT seems more like the properties of the proposed method, rather than a framework.
>
> CAT is only our first concrete instantiation; we do not claim it is the only one.
> In fact, the EIT definition is satisfied by many other transformations.
> For example, a simple average pooling (or any learnable weighted sum) along the value stream already meets all four EIT criteria.
> Likewise, adding a constant bias or multiplying with Hadamard product before the CAT's softmax + gather operation still produces an EIT's layer, because the exact row‑wise softmax is preserved.
> Beyond circular convolutions, other Toeplitz‑structured transforms can also be devised.
>
> Put differently, once we insist on exact softmax but allow any sub‑quadratic implementation, the design space becomes as rich as the kernel choices explored by the LinearAttention family after they drop the softmax constraint. We therefore view CAT as only the first step; we expect
> future work, either variants of CAT or entirely different constructions, to populate the broader EIT family.
>
> > Moreover, I do not understand why “Softmax preservation” is a property that is a priori ideal.
>
> We do not claim that exact softmax is universally optimal; efficiency, accuracy, and solid theory ultimately decide. Still, keeping softmax is worth exploring:
> - LinearAttention methods begin by discarding softmax, then explore a huge kernel space. EITs take the mirror position: we keep softmax but allow any sub‑quadratic implementation. The two lines of work are therefore complementary. exploring only one side leaves the other side under‑investigated and risks missing the sweet spot if exact soft‑max should turn out to be important.
> - Our CAT layer, which preserves softmax, is faster than baseline while matching or improving accuracy. This shows the softmax branch can be both efficient and competitive.
>
> Thus, soft‑max preservation is not an article of faith, but a plausible design axis that deserves systematic investigation; CAT is simply the first data point on that axis.
>
>
> 2. The proposed CAT is not well motivated and justified.
>
> Thank you for highlighting the role of permutation equivariance.
> Its importance strongly depends on the domain, so we address the issue by task category below.
>
> 2.1 Tasks where order carries meaning (language, vision, etc.)
>
> In these domains, permutation equivariance is already broken on purpose by positional encodings, because absolute or relative position improves accuracy. While cat retains cyclic-shift equivariance, a restricted form of permutation equivariance in which the output permutes correctly under any circular shift of the input, CAT therefore operates under exactly the same assumptions as standard self-attention.
>
> 2.2 Tasks where order is undefined or irrelevant (sets, graphs, particle systems)
>
> For inputs that are true sets, full permutation equivariance is essential. The current form of CAT does not satisfy this requirement. However, two clear research avenues exist:
>
>     i) Explicit order prediction — adopt a learnable sorting layer (e.g., SoftSort ) and re-order the output to match the input exactly.
>     ii) Hybrid models – combine CAT with an order-assigning encoder (e.g., self-attention equipped with a positional encoding) so that CAT captures local cyclic patterns within that imposed order, while a subsequent permutation invariant read-out aggregates the final set representation.
>
> Exploring these directions is outside the scope of this paper but, in our view, represents a promising next step for applying CAT-style efficiency to truly order-agnostic data.
>
> 3. The efficiency claim of the third contribution lacks detailed evidence.
>
> Please refer to the profile table (V100, FP16, Adam, Batchsize = 32):
> torch2.7.0+cu126+cudnn9.5.1 default's cuFFT(torch.fft) / FlashAttention(F.scaled_dot_product_attention) used.
> For each entry we report the average over 10 consecutive steps, with the runtime decomposed into three parts:
> “fwd” = forward pass, “bwd” = backward pass, and “opt” = optimizer update.
>
> | CLIP-L Variant                    | Param | Peak mem | Average time (1 iter.)|
> |-----------------------------|---------:|--------:|------------:|
> | Self‑Attn (PyTorch MatMul)     | 304.2M | 16654 MB | 1465.72 ms (fwd: 490.05 / bwd: 940.28 / opt: 35.39) |
> | Self‑Attn (FlashAttention)     | 304.2M | 13594 MB | 1467.50 ms (fwd: 465.77 / bwd: 966.46 / opt: 35.27) |
> | CAT (gather, O (N²))        | **254.2 M** | 14931 MB | 1188.48 ms (fwd: 401.12 / bwd: 757.52 / opt: 29.84) |
> | CAT (cuFFT, O (N log N))    | **254.2 M** |**11 882 MB** | **1187.85 ms** (fwd: 401.34 / bwd: 756.63 / opt: 29.88)|
>
> | GPT2-small Variant                    | Param | Peak mem | Average time (1 iter.)|
> |-----------------------------|---------:|---------:|------------:|
> | Self‑Attn (PyTorch MM)      | 107.5 M |5289.1MB | 521.57  ms (fwd: 168.99 / bwd: 339.69 / opt: 12.88) |
> | Self‑Attn (FlashAttention)     | 107.5 M |  5258.5MB | 531.05 ms (fwd: 164.75 / bwd: 350.50 / opt: 12.75) |
> | CAT (gather, O (N²))        | **93.5 M** | **4651.3 MB** | **441.78 ms** (fwd: 114.37 / bwd: 286.17 / opt: 11.24) |
>
> We use cuFFT as the FFT backend. As the table shows, cuFFT is both fast and memory-efficient in its standard configuration, but we observed two practical issues that leave room for improvement:
> - Limited mixed-precision support — Performance drops noticeably for sequence lengths that are not powers of two.
> - High overhead for short sequences — For small N, the kernel launch and planning cost dominates, so the non-FFT (gather) version of CAT often runs faster than the cuFFT-based implementation.
>
> We also checked fft‑conv‑pytorch, but it supports only normal convolutions and cannot handle the 4‑D circulant tensors required by CAT. Future GPUs with faster FFT kernels can furtherimprove the O (N log N) path.
>
>
> If any of our replies are unclear or leave questions unanswered, please feel free to add further comments or adjust your evaluation. Once again, thank you very much for your thoughtful review!

---

> ### Comment · Reviewer_UTRX · 2025-08-07
> **Thanks to the authors for their clarification**
>
> Thank you for the detailed clarification! It has addressed my concerns. I encourage the authors to integrate them in a revised version of the paper. I have increased my rating accordingly.

---

### Note · Authors · 2025-08-15

We are deeply grateful to the reviewers and the AC for the exceptionally thoughtful feedback and discussion.
Our work targets a question: can we retain exact softmax while reducing complexity below O(N^2)? CAT concretely demonstrates that exact softmax can coexist with sub-quadratic efficiency, achieving O(N log N) via FFT with no hyper-parameter, reduced learnable parameters, and about 10% speedups. It requires no parameter tuning, interoperates with Flash-style kernels and GQA, and matches or improves accuracy on ImageNet-1k and WikiText-103.

Following the discussion, we clarified:
- Scope of EITs. EITs are not tied to CAT; we outlined additional candidates and design space beyond circular convolutions.
- Efficiency evidence. We provided wall-clock and memory profiles on V100/FP16/B=32. CAT (gather) is about 10% faster than the baseline in our setting, and the cuFFT variant reduces peak memory while remaining competitive.
- Compatibility. CAT is compatible with efficient kernels (e.g., Flash-style methods ) and with GQA; small runs indicate CAT-alter is at least competitive.
- Permutation equivariance. Standard practice of self-attentions adds positional encodings, thereby breaking global permutation equivariance; while CAT (without positional encodings) is cyclic-shift equivariant, the CAT with positional encodings operates under the same assumption and no longer preserves that symmetry.
- Limitations and future work. Truly ultra-long causal contexts and fully permutation-equivariant variants remain open; we outlined concrete next steps (efficient FFT kernels, hybrid/equivariant extensions).

Since submission, analyses of softmax expressiveness (e.g., recent RNN-perspective softmax work) and application-driven directions in softmax self-attention architectures further underscore the relevance of exploring this “keep softmax, make it efficient” design axis. Regardless of the final decision, we believe the clarifications prompted by your comments on profiling, scope, and limitations will help future work examine this space more rigorously (including hybrid and equivariant variants).

We sincerely thank the reviewers and the AC for the time and thoughtful feedback; we are confident it has strengthened the clarity and framing of this line of work going forward!

---

### Decision · Program_Chairs · 2025-09-17

**Decision:**

Accept (poster)

**Comment:**

The paper proposes an alternative to standard attention in Transformers that is based on circular convolutions and allows efficient speedup through Fourier domain computation, reducing complexity to be n log n instead of quadratic in sequence length.

While the final scores stand at accept, borderline accept, and borderline reject, all reviewers acknowledge the merit of the idea and work. Overall, the rebuttal added experimental evaluations that helped ameliorate many of the reviewers' concerns. While two of the reviewers are positive, the borderline negative reviewer notes that the paper may need more experimental results to fully resolve their concerns.

After considering all reviews and the rebuttal, AC believes that while additional results could add value to the paper, they are not strictly necessary. The novelty of the method and the existing results (in the paper and the rebuttal) are sufficient to clear the bar for acceptance, and the work is likely to be of considerable interest to the NeurIPS audience.